# A cascade training protocol for schoolteachers on oral health: Investigating the influence of primary and secondary mentors

**Sarah Paul**[1�} , **Neha Chauhan**[1☉], **Ritu Duggal**[2☉], **Bharati Purohit**[3☉], **Harsh Priya**[3☉] *

**1** National Oral Health Program, Centre for Dental Education and Research, All India Institute of Medical Sciences, New Delhi, India, **2** All India Institute of Medical Sciences, Centre for Dental Education and Research, New Delhi, India, **3** Department of Public Health Dentistry, All India Institute of Medical Sciences, Centre for Dental Education and Research, New Delhi, India

☉ These authors contributed equally to this work.
* drharshpriya@gmail.com

**Data Availability Statement:** The data will be managed by the Institute Ethics Committee of All India Institute of Medical Sciences (AIIMS) New Delhi. For access to the data, please contact the

## Abstract

### Background

Oral health holds paramount importance for overall well-being, particularly among school-aged children, where inadequate oral health can cause significant discomfort and impede educational performance. Despite its critical nature, oral health awareness and practices within Indian school communities remain suboptimal. There exists an urgent necessity for efficacious educational initiatives to bridge this gap and foster oral health awareness among both schoolteachers and students.

### Protocol

This monitored training introduces a pioneering cascade model training initiative aimed at evaluating its impact on enhancing oral health knowledge among schoolteachers and students in Indian schools. A total of 100 school teachers from 50 Centre government schools will undergo training in two distinct sessions. Following this, two teachers from each school will be selected to further train 24 class monitors from grades 6–9, who will subsequently impart knowledge to approximately 40 students per class. This phase of the training will be ongoing, ensuring training consistency through online supervision and active participation of the expert team in training activities with newly designated master trainers. Pre- and post-training assessments will be conducted at each level to gauge the monitored training's effectiveness. The entire training is projected to conclude within a year.

### Discussion

This cascade model monitored training initiative represents a novel approach in promoting oral health awareness in Indian schools, marking a pioneering endeavor in this domain. Through this program, we aim to empower 1200 class monitors as master trainers and

ethics committee using the following email Id
ethicscommitteeaiims@gmail.com.

**Funding:** The author(s) received no specific funding for this work.

**Competing interests:** The authors have declared that no competing interests exist.

reach approximately 24,000 school students across 480 supervised training sessions. The establishment of master trainers through a cascade process, coupled with the engagement of the CDER expert team, ensures accurate dissemination of information at every stage. The comprehensive evaluation facilitated by pre- and post-training assessments at each level further enhances the program's effectiveness, laying a solid foundation for future oral health initiatives within school communities.

## Introduction

Oral health is integral to overall well-being, significantly impacting an individual's ability to communicate, eat, and socialize comfortably, especially during childhood. However, despite its importance, oral health issues among children, such as dental problems leading to discomfort and school absences, remain prevalent [1].

The profound impact of poor oral health on educational outcomes cannot be understated. Children experiencing dental pain or discomfort are more likely to miss school days, leading to significant academic setbacks. Additionally, dental issues can hinder concentration in class, diminishing learning efficiency and academic performance. Addressing oral health concerns within school settings is crucial for the physical health, educational success, and long-term well-being of students [2].

Schools serve as crucial platforms for promoting oral health, providing access to a diverse population of children. Teachers, as frontline caregivers, play a pivotal role in identifying oral health issues among students and promoting preventive measures. However, many teachers lack the necessary training and resources to effectively address oral health concerns [3].

This underscores the need for comprehensive teacher training programs focused on oral health promotion. By equipping teachers with the knowledge and skills to educate students about oral hygiene practices, these programs empower educators to become agents of change within their classrooms and school communities. Leveraging the Cascade model of professional training, these programs aim to create a ripple effect, producing master trainers among both teachers and students.

### Rationale of the training program

Existing literature demonstrates the effectiveness of school-based interventions in improving oral health outcomes among children, particularly in low- and middle-income countries [4] Furthermore, training programs targeting teachers have shown promising results in enhancing their knowledge and promoting positive oral health behaviours among students [5].

In this context, our training program aims to go beyond merely training teachers; we strive to create a cascade effect by producing master trainers among both teachers and students. Utilizing the Cascade model of professional training, our program equips teachers with the necessary knowledge and skills to train monitors within their classrooms [6]. These monitors, in turn, become adept at disseminating oral health education to their peers, effectively becoming secondary trainers themselves.

Emphasizing the development of master trainers among both teachers and students, our initiative seeks to create a sustainable model of oral health promotion within school settings. Through a rigorous evaluation of the training program's outcomes, we aim to generate evidence of its effectiveness for large-scale implementation. Ultimately, our goal is to foster a

culture of oral health awareness and responsibility among students and teachers, contributing to healthier and more successful learning environments [7,8].

### Study design

This training employs an interventional cascade model to assess the effectiveness of a training program on oral health promotion among schoolteachers, class monitors, and students. The cascade model involves training school teachers, who then train class monitors, who subsequently educate their classmates on oral health practices.

### Sample size estimation

Participants were nominated by a central educational institution, ensuring diversity in background characteristics. A total of 100 schoolteachers were selected from different regions of India. Each school teacher will further train 24 class monitors from classes 6 to 9, who will then educate approximately 40 students each. Administrative approvals have been obtained for the study. Fig 1 illustrates the total sample size covered in the study, encompassing school teachers, class monitors, and students from various regions of India. The sample includes participants from different schools nationwide, selected by the central educational institution.

### Sampling strategy

School teachers were nominated by the central educational institution, representing various regions of India. Class monitors and their respective classes for further training will be selected by the participating school teachers.

### Inclusion criteria

Participants include schoolteachers, class monitors, and high school students selected from various schools across five zones of India by the central educational institution.

### Exclusion criteria

Individuals unwilling to participate or provide informed consent will be excluded from the study.

### Training instrument

A pre-tested, validated, closed-ended, self-structured questionnaire on oral health knowledge, attitude, approach, and actions change will be designed in English for data collection

### Methodology

Ethics approval for this study has been taken from the Institute Ethics Committee of All India Institute of Medical Science (Ref. No.: IEC- 210/16.05.2023, RP- 16/2023). Additionally, blanket approval has been secured from the central regulating body of participating government institutions and their respective regional headquarters. The study has been duly registered with the Clinical Trial Registry of India (CTRI) under the reference number REF/2023/05/066717. Informed consent will be obtained from teachers, students, and parents of participating students, ensuring their full understanding and voluntary participation in the study.

The 100 school teachers from 50 schools (2 each) from 5 different zones in India will be selected by the central educational institution, divided into 2 batches containing 50

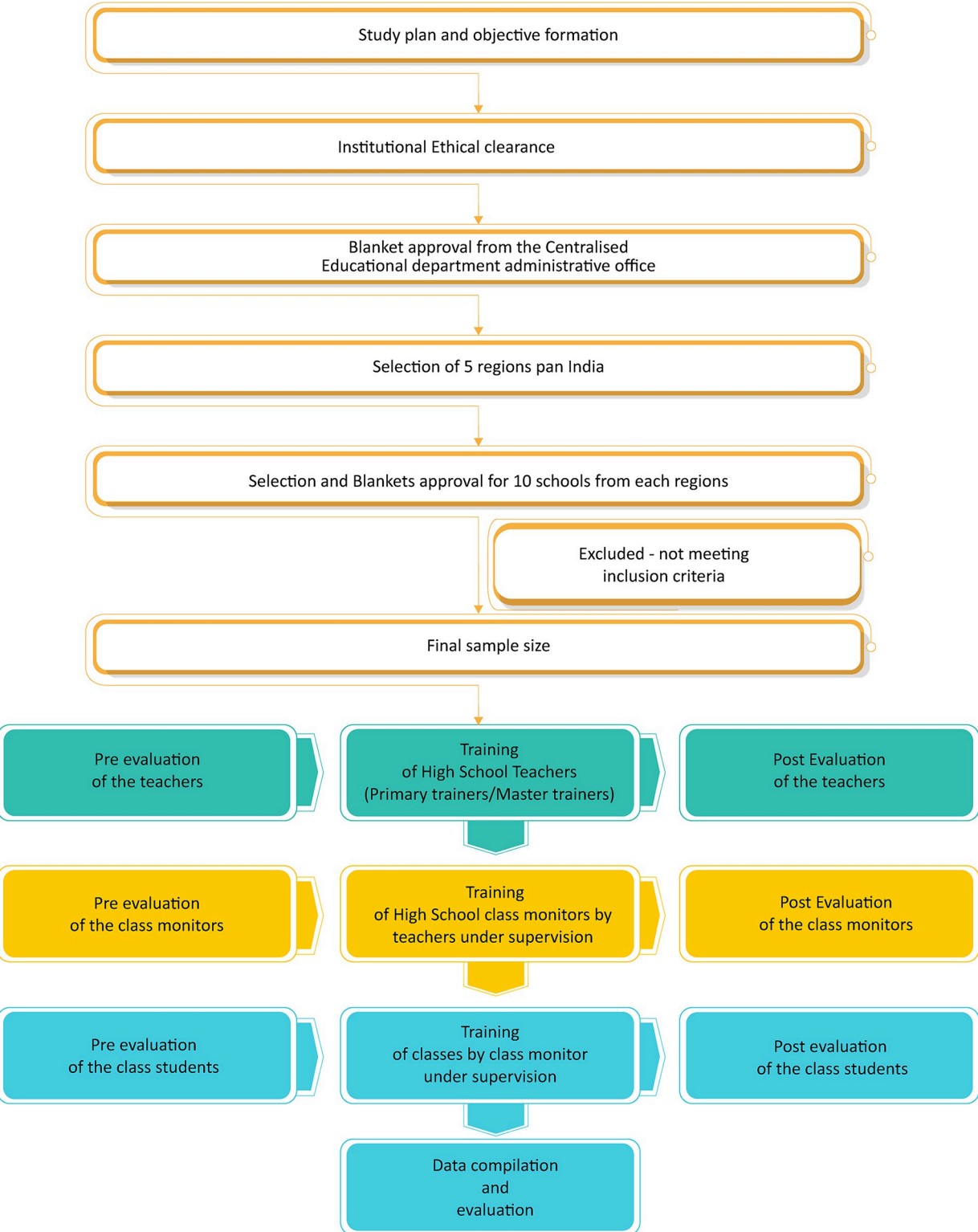

**Fig 1. Representation of the total sample size covered for the study, including school teachers, class monitors, and school students from various regions of India.** The sample size is inclusive of participants selected from different schools across India by the central educational institution.

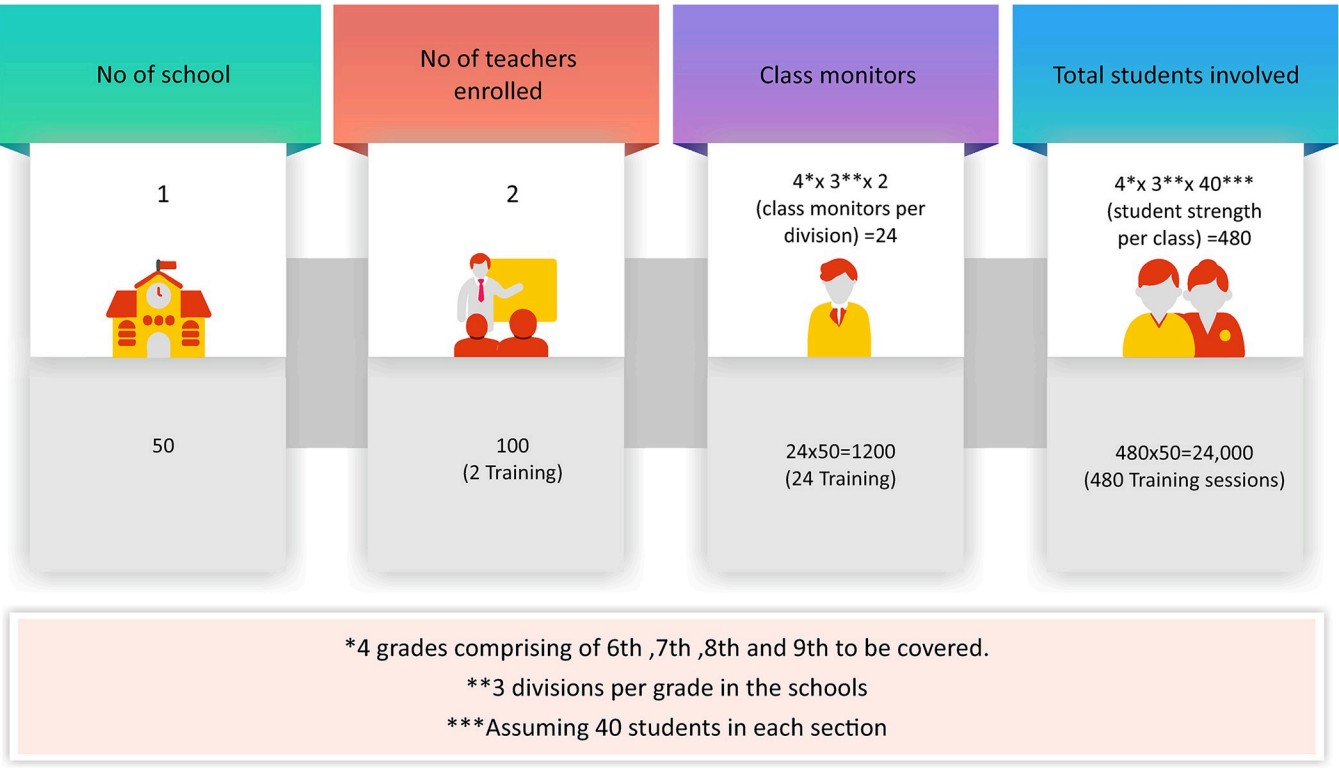

**Fig 2. Flow chart of the methodology of the Cascade progression among the three different tiers.**

schoolteachers each. Each batch will attend 1 session of training. (Fig 2) The training module will be conducted using an online webinar series comprising standardized PowerPoint presentations.

Prior to the training, all participants will complete a pre-test in the form of an online semi-structured self-administered questionnaire (in English) using Google Documents. After training, post-training data will be collected using the same online questionnaire. A Google feedback form will also be provided at the end of the training session to gather recommendations for program improvement.

Each teacher who underwent training from a particular school, i.e., primary trainers, will be assigned to train 24 class monitors each from grades 6th to 9th. This training will be monitored online by the CDER team, with sessions divided over 24 working days. All class monitors will complete a pre-test using an online semi-structured self-administered questionnaire.

Subsequently, the 2 class monitors of each class will be assigned to train their respective class of 40 students each again monitored by the CDER team. Pre- and post-training data of each student will be obtained using questionnaires shared via Google Forms. It will be ensured that each student undergoing training has access to a computer.

## Monitoring

Monitoring will follow a structured format, including capacity building through training of teachers, inculcating oral hygiene practices among students, and reaching out to a larger number of people through the cascade model.

## Statistical analysis

The data will be analyzed using descriptive statistics to summarize participant demographics and baseline oral health metrics. Paired t-tests or Wilcoxon signed-rank tests will compare pre- and post-training scores on oral health knowledge, attitudes, approaches, and actions. Logistic regression or generalized linear mixed models will identify factors associated with improved oral health outcomes. Subgroup analyses will assess differential effects based on demographics or school settings, with interaction effects explored. Correlation analyses will assess associations between changes in oral health metrics. Statistical significance will be set at $p < 0.05$, and analyses will be conducted using IBM SPSS Statistics version 27. Qualitative feedback from the Google form will complement quantitative findings to assess program effectiveness and identify areas for improvement.

## Outcome measures

The impact of oral health promotion training for school teachers on knowledge, attitude, approach, and actions change will be assessed. The effectiveness of the training program will be evaluated, along with recommendations to enhance the oral health training program based on discussions with school teachers.

## Primary outcome measures

1. Existing knowledge, attitude, approach, and actions of school teachers on oral health.

2. Effectiveness of the training program.

3. Recommendations to escalate the oral health training program based on views and discussions with school teachers.

## Secondary outcome measures

1. Existing knowledge, attitude, and actions of high school students regarding oral health.

2. Effectiveness of the cascade model of information transmission in oral hygiene promotion.

## Uniqueness of the study

This monitored training presents a unique approach to oral health promotion within school settings through the implementation of an interventional cascade model. Several aspects contribute to the uniqueness of this study:

Cascade Model Implementation: This training utilizes a cascade model, wherein school-teachers are trained to become primary trainers who then train class monitors, ultimately leading to the education of students. This cascading approach ensures the dissemination of oral health knowledge across multiple layers within the school environment, maximizing its reach and impact.

Integration of Technology: This training leverages online webinar series and digital platforms such as Google Documents and Forms for training delivery, data collection, and feedback mechanisms. This integration of technology enables efficient and scalable implementation of the training program, especially in the context of diverse geographic locations.

Collaboration with Central Educational Institution: Collaboration with a central educational institution ensures the involvement of a diverse pool of participants from various

regions of India. This collaboration enhances the generalizability of findings and facilitates the implementation of findings at a national level.

Focus on Capacity Building: This training emphasizes capacity building not only among schoolteachers but also among class monitors and students. By empowering multiple stakeholders within the school community, the training aims to create a sustainable culture of oral health awareness and responsibility.

Comprehensive Outcome Measures: The training evaluates a wide range of outcome measures, including changes in knowledge, attitudes, approaches, and actions among schoolteachers and students. This comprehensive assessment provides valuable insights into the effectiveness of the training program and its impact on oral health promotion within schools.

Potential for Scalability: The study's focus on developing a scalable training module at a renowned institution like AIIMS suggests potential for nationwide and even worldwide implementation of similar programs. By establishing a baseline for future oral health promotion initiatives, the training contributes to the advancement of public health efforts on a global scale.

## Possible limitation and challenges

Ensuring active participation and engagement from all participants, including school teachers, class monitors, and students, may be challenging. Resistance to change or competing priorities could hinder the uptake of new oral health practices and concepts, undermining the program's success.

Furthermore, sustaining momentum beyond the initial training phase presents a significant challenge. Without ongoing support and reinforcement, there is a risk that the knowledge and skills gained during the training may not be sustained over time. Cultural and linguistic diversity among participants also poses a challenge, requiring tailored approaches to ensure inclusivity and effectiveness. Measurement and evaluation of the program's impact may be methodologically challenging, and logistical considerations such as coordinating training sessions across multiple schools and regions require careful planning. Overcoming resistance to change and fostering a culture of acceptance and commitment to oral health promotion are essential for the success of the training program.

## Current status

### Pilot study

The initial patient recruitment occurred in the form of requesting teachers to complete a pre-training questionnaire, followed by a training session in the first week of November 2023. These activities transpired after the receipt of ethics approval on 26 May 2023. Subsequently, two teachers from this training further disseminated the training to their monitors and class students, with data collection commencing in January 2024 which we used for pilot analysis. A pilot study was undertaken to assess the effectiveness of a cascade training approach. Here are the findings regarding the mean scores of oral health knowledge before and after the training (Table 1).

- Among teachers, there was a slight increase in the mean oral health knowledge score after the training.

- Class monitors showed a significant improvement in their mean oral health knowledge score after the training ($p < 0.001$).

- Similarly, students demonstrated a significant enhancement in their mean oral health knowledge score after the training ($p < 0.001$).

**Table 1. The table outlining the mean pre-training scores, mean post-training scores, and associated p-values for teachers, class monitors, and students.**

|  | MEAN PRE-Questionnaire | MEAN POST Questionnaire | P VALUE |
|---|---|---|---|
| Teachers (n = 2) | 13.5 | 14.5 | 0.5 |
| Class monitors (n = 15) | 8.53 ± 2.53 | 12 ± 2.07 | >0.001 |
| Students (n = 187) | 8.10 ± 2.51 | 10.58 ±2.26 | >0.001 |

These results indicate a positive impact of the cascade training program on oral health knowledge, particularly notable among class monitors and students. It's important to note that the lack of statistical significance in the teachers' group may be due to the small sample size (n = 2) and the primary focus of the training on knowledge dissemination to monitors and students. Overall, these findings underscore the effectiveness of the training program in promoting oral health awareness within the school community.

## Status of the current study

The training has advanced to the second stage, with the completion of initial training for school teachers. Currently, focus is on training class monitors in the cascade model. The first stage, training school teachers, has concluded successfully, setting the groundwork for further dissemination of oral health education within school communities. Transitioning to the second stage marks a milestone in implementing the cascade model, fostering sustainable oral health promotion within schools

## Conclusion

In conclusion, this monitored training initiative presents a novel and comprehensive approach to oral health promotion within school settings. By implementing an interventional cascade model and leveraging technology-enabled training delivery, the initiative aims to empower schoolteachers, class monitors, and students with the knowledge and skills necessary to promote oral health practices effectively.

Through collaboration with a central educational institution and the involvement of participants from diverse geographic regions, the initiative seeks to generate findings that are not only applicable at a local level but also have the potential for national and global scalability. By focusing on capacity building and evaluating a wide range of outcome measures, including changes in knowledge, attitudes, approaches, and actions, the initiative aims to contribute valuable insights to the field of oral health promotion.

Ultimately, the success of this monitored training initiative has the potential to establish a sustainable culture of oral health awareness and responsibility within schools, thereby improving the oral health outcomes of children and adolescents. Furthermore, by serving as a baseline for future oral health promotion initiatives, the initiative paves the way for the advancement of public health efforts on a broader scale.

In summary, this monitored training initiative represents a significant step towards addressing oral health disparities and promoting overall well-being among school-aged populations, highlighting the importance of holistic approaches to health promotion within educational settings.

## Author Contributions

**Conceptualization:** Sarah Paul, Neha Chauhan, Bharati Purohit, Harsh Priya.

**Data curation:** Sarah Paul, Neha Chauhan.

**Formal analysis:** Sarah Paul.

**Investigation:** Sarah Paul, Neha Chauhan, Bharati Purohit, Harsh Priya.

**Methodology:** Sarah Paul, Bharati Purohit, Harsh Priya.

**Project administration:** Sarah Paul, Neha Chauhan, Bharati Purohit, Harsh Priya.

**Supervision:** Ritu Duggal, Bharati Purohit, Harsh Priya.

**Validation:** Sarah Paul, Neha Chauhan, Ritu Duggal, Bharati Purohit, Harsh Priya.

**Visualization:** Sarah Paul, Neha Chauhan, Ritu Duggal, Harsh Priya.

**Writing – original draft:** Sarah Paul.

**Writing – review & editing:** Neha Chauhan, Ritu Duggal, Bharati Purohit, Harsh Priya.

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
