## [Decision Letter · Decision Letter 0]

9 Jul 2024

PONE-D-24-17241A Cascade Training Protocol for School Teachers on Oral Health: Investigating the Influence of Primary and Secondary Mentors.PLOS ONE

Dear Dr. Priya,

Thank you for submitting your manuscript to PLOS ONE. After careful consideration, we feel that it has merit but does not fully meet PLOS ONE’s publication criteria as it currently stands. Therefore, we invite you to submit a revised version of the manuscript that addresses the points raised during the review process. 

We look forward to receiving your revised manuscript.

Kind regards,

Apurva kumar Pandya, PhD

Academic Editor

PLOS ONE

2. In the online submission form, you indicated that [Individual de-identified participant data will be made available on reasonable request, from the corresponding author , starting from the date of publication, until 10 years after publication. Requests beyond this timeframe will be considered on a case-by-case basis. In addition, the study protocol, including the statistical plan are already available as a supplementary appendix attached to this manuscript]. 

Additional Editor Comments:

The protocol is well written. I agree with the reviewers' suggestion to make minor changes to the manuscript.

Reviewers' comments:

Reviewer's Responses to Questions

**Comments to the Author**

1. Does the manuscript provide a valid rationale for the proposed study, with clearly identified and justified research questions?

Reviewer #1: Yes

2. Is the protocol technically sound and planned in a manner that will lead to a meaningful outcome and allow testing the stated hypotheses?

Reviewer #1: Yes

3. Is the methodology feasible and described in sufficient detail to allow the work to be replicable?

Reviewer #1: Yes

4. Have the authors described where all data underlying the findings will be made available when the study is complete?

Reviewer #1: Yes

5. Is the manuscript presented in an intelligible fashion and written in standard English?

Reviewer #1: Yes

6. Review Comments to the Author

You may also provide optional suggestions and comments to authors that they might find helpful in planning their study.

Reviewer #1: The authors should explain what educational tool they have used to educate the school teachers and the students regarding oral health care. The authors can also add the details of Pre- and post-training assessments they have used to assess the knowledge gain among teachers and students.

7. PLOS authors have the option to publish the peer review history of their article (what does this mean?). If published, this will include your full peer review and any attached files.

Reviewer #1: **Yes: **Dr Girish Suragimath, Professor and Head, Department of Periodontology, School of Dental Sciences, Krishna Vishwa Vidyapeeth, Karad, Maharshtra, India.

---

## [Author Response · Author response to Decision Letter 0]

5 Aug 2024

Dear Dr. Apurva Pandya,

Thank you for your valuable feedback on our manuscript. We have addressed the points raised in your review as follows:

Suggestion 1: Formatting and Style Requirements

Action Taken: We have ensured that our manuscript meets the PLOS ONE style requirements, including those for file naming. The revised manuscript adheres to the guidelines provided in the PLOS ONE style templates.

Suggestion 2: Data Availability

Action Taken: In accordance with PLOS ONE's data availability policy, we confirm that de-identified participant data will be made available on reasonable request from the corresponding author. The data underlying our findings will be deposited in a public data repository and made accessible in a masked data sheet to ensure participant confidentiality.

Suggestion 3: Reference List Review

Action Taken: We have reviewed our reference list thoroughly to ensure it is complete and correct. We did not find any retracted papers among our citations. Therefore, no retractions are cited in our manuscript, and all references are up-to-date and relevant.

Thank you for your consideration. We look forward to your positive response.

Kind regards,

Dr. Harsh Priya, 

Additional Professor, 

Department of Public Health Dentistry

Centre for Dental Education and Research, 

All India Institute of Medical Sciences, New Delhi.

---

## [Editor Report · Decision Letter 1]

16 Aug 2024

A Cascade Training Protocol for School Teachers on Oral Health: Investigating the Influence of Primary and Secondary Mentors.

PONE-D-24-17241R1

Dear Dr. Priya,

We’re pleased to inform you that your manuscript has been judged scientifically suitable for publication and will be formally accepted for publication once it meets all outstanding technical requirements.

Kind regards,

Apurva kumar Pandya, PhD

Academic Editor

PLOS ONE
---

## [Editor Report · Acceptance letter]

21 Aug 2024

PONE-D-24-17241R1 

PLOS ONE

Dear Dr. Priya, 

I'm pleased to inform you that your manuscript has been deemed suitable for publication in PLOS ONE. Congratulations! Your manuscript is now being handed over to our production team.

Kind regards, 

on behalf of

Dr. Apurva kumar Pandya 

Academic Editor

PLOS ONE